# Post-Ischemic Stroke Cardiovascular Risk Prevention and Management

**DOI:** 10.3390/healthcare12141415

**Published:** 2024-07-16

**Authors:** Yilei Guo, Danping Pan, Haitong Wan, Jiehong Yang

**Affiliations:** 1College of Basic Medical Sciences, Zhejiang Chinese Medical University, Hangzhou 310053, China; guoyilei86@163.com (Y.G.); 13503827697@163.com (D.P.); 2The First Affiliated Hospital, Zhejiang Chinese Medical University, Hangzhou 310003, China; whtong@126.com; 3Institute of Cardio-Cerebrovascular Disease, Zhejiang Chinese Medical University, Hangzhou 310053, China; 4Key Laboratory of TCM Encephalopathy of Zhejiang Province, Hangzhou 310053, China

**Keywords:** ischemic stroke, cardiovascular disease, adverse cardiovascular events, risk management

## Abstract

Cardiac death is the second most common cause of death among patients with acute ischemic stroke (IS), following neurological death resulting directly from acute IS. Risk prediction models and screening tools including electrocardiograms can assess the risk of adverse cardiovascular events after IS. Prolonged heart rate monitoring and early anticoagulation therapy benefit patients with a higher risk of adverse events, especially stroke patients with atrial fibrillation. IS and cardiovascular diseases have similar risk factors which, if optimally managed, may reduce the incidence of recurrent stroke and other major cardiovascular adverse events. Comprehensive risk management emphasizes a healthy lifestyle and medication therapy, especially lipid-lowering, glucose-lowering, and blood pressure-lowering drugs. Although antiplatelet and anticoagulation therapy are preferred to prevent cardiovascular events after IS, a balance between preventing recurrent stroke and secondary bleeding should be maintained. Optimization of early rehabilitation care comprises continuous care across environments thus improving the prognosis of stroke survivors.

## 1. Introduction

Stroke stands as a primary cause of mortality and disability worldwide [1]. Ischemic stroke (IS) is the most significant challenge contributing to this burden [1]. Compared with patients without a history of stroke or with hemorrhagic stroke, patients with IS have a higher risk of adverse cardiac events [2,3]. These adverse cardiac events greatly exacerbate the mortality of patients with IS [4]. According to reports, patients with acute ischemic stroke (AIS), who subsequently experience acute myocardial infarction (MI), face a threefold increased risk of in-hospital mortality compared with those who do not experience acute MI [4]. Currently, among individuals with AIS, cardiac death has emerged as the second leading cause of death, surpassed only by fatalities directly linked to the neurological damage from stroke itself [5], highlighting the importance of preventing adverse cardiovascular events after IS. Therefore, optimizing prevention and management to preclude or delay the occurrence of recurrent stroke and other cardiovascular events is an urgent issue. We believe that secondary prevention and post-discharge rehabilitation care should be emphasized for patients with a history of IS.

In 1947, Byer et al. [6] recognized that cerebrovascular diseases can result in myocardial injury and arrhythmias, marking a significant milestone in our understanding of the interplay between cerebrovascular and cardiovascular health. Some scholars have conceptualized post-stroke cardiac events as ‘stroke–heart syndrome’, aiming to refine clinical insights and enhance the management of such conditions [7]. This concept implies that cardiac dysfunction occurs after the onset of neurological deficits. AIS can cause cardiac dysfunction through pathways such as sympathetic and parasympathetic imbalance, catecholamine surge, immune and inflammatory responses, and gut microbiota dysbiosis [8]. Presently, most studies on stroke–heart syndrome primarily focus on its clinical presentations and pathophysiological underpinnings, with a relatively scant focus on preventive management strategies for post-IS heart disease.

In this review, first, we briefly outline the epidemiological characteristics of cardiovascular events after IS. Subsequently, we review the risk factors and risk prediction scores for adverse cardiovascular events after IS. For patients with a higher risk of adverse cardiovascular events, prolonged heart rate monitoring time and early anticoagulation therapy can bring benefits. Next, we focus on the management strategies for modifiable risk factors and the preferred treatment methods for preventing cardiovascular disease (CVD) after stroke. Additionally, we briefly introduce measures to promote continuity of care in different healthcare environments in order to optimize early rehabilitation care for patients with IS and improve their overall prognosis. These strategies involve various disciplines and call for multidisciplinary team collaboration. Finally, we propose ideas for future research aimed at providing directions for improving the clinical management of CVDs after stroke. 

## 2. Epidemiological Characteristics of Post-IS Cardiovascular Events

After AIS, CVD risk increases over both the short term and long term. A prospective study reported that during the acute or short-term phase, 25.1% of patients with AIS developed significant arrhythmias within the initial 72 h of hospitalization; this probability peaked within the first 24 h and gradually declined thereafter [9]. A retrospective cohort study found that within 4 weeks after IS, 11.1% of patients developed acute coronary syndrome, 8.8% experienced atrial fibrillation (AF) or flutter, 6.4% had heart failure, 1.2% had severe ventricular arrhythmias, and 0.1% had Takotsubo syndrome [10]. Within 90 days post-IS, 0.26% of patients were readmitted due to venous thromboembolism. The risk of readmission was highest within the first 4 to 6 weeks [11]. Notably, among 47,229 patients with IS, 3% continued to be at risk of MI in the 1-year post-IS period [12]. It was observed that the risk of cardiovascular events within 1 year after AIS was highest in the first 30 days but declined both during the 31–90-day and 91–365-day periods [3]. We can regard cardiovascular events occurring within 30 days of IS as “stroke–heart syndrome”. Once outside of this time window, cardiovascular events are defined as possible long-term complications because their correlation with stroke is weaker [3].

Myocardial damage following an acute stroke may be influenced by individual myocardial vulnerability owing to cardiac factors, such as pre-existing structural heart disease [13], or it may be attributed to autonomic and inflammatory mechanisms mediated by damage to the brain–cardiac axis [14], or both. When stroke acts as an independent risk factor for CVDs, it may, along with other risk factors such as hyperlipidemia, hyperglycemia, and hypertension, increase the risk of recurrent stroke and cardiovascular events. Stroke may also indicate underlying cardiac defects or cardiovascular issues such as valvular heart disease or AF, which may underlie or coexist with stroke. Insular cortex lesions can help differentiate cardioembolic stroke (caused by AF) from non-cardioembolic stroke (caused by atherosclerosis or cerebral microvascular disease) and are considered potential neuroimaging markers for identifying IS subtypes due to cardiac embolism [15]. Even in patients without known preexisting cardiac comorbidities, the first-ever IS is independently associated with an increased risk of major adverse cardiovascular events, including MI, unstable angina, congestive heart failure, and coronary artery disease [3]. A comprehensive meta-analysis revealed that as many as one-third of patients with AIS without a history of cardiac disease exhibited coronary artery stenosis exceeding 50% [12]. Additionally, it is crucial to note that cerebral blood flow regulation during AIS is directly dependent on the heart rather than on the brain [16]. Consequently, adverse cardiac events following AIS can further exacerbate brain damage.

As patients with IS face a high risk of adverse cardiovascular events, treatment for post-stroke cardiovascular complications leads to significantly increased indirect costs. It was estimated that the total direct and indirect economic losses attributable to stroke in 2017 was nearly US $900 billion worldwide [17]. Given the enormous indirect costs associated with adverse cardiovascular events, preventive interventions targeting individual risk factors may be the most cost-effective choice [18]. Evidence suggests that every dollar spent on stroke and CVD prevention results in a return on investment of over 10 dollars [18].

## 3. Risk Stratification

According to published guidelines, common cardiovascular risk factors include a sedentary lifestyle, smoking, being overweight, unhealthy dietary habits, diabetes, and hypertension [19,20]. Notably, many studies have explored the risk factors for CVDs; however, there has been relatively little focus on the specific risk management for CVDs following IS. Some population-based sample studies have achieved success in further investigating the risk factors for CVDs after IS. By analyzing the National Inpatient Sample between 2003 and 2014 (AIS = 864,043), Alqahtani et al. found that older age, history of coronary artery disease, chronic renal insufficiency, undergoing mechanical thrombectomy, and rhythm and conduction abnormalities were the strongest predictors of developing AMI after AIS [4]. Recent studies have demonstrated that reduced albumin levels (≤3.4 g/dL) [21] and triglyceride–glucose index (≥8.9) [22] are also predictors of cardiovascular complications subsequent to IS.

Risk prediction scores or models, in conjunction with screening tools such as electrocardiography (ECG), can help identify high-risk patients who can then benefit from early risk management and guided preventive treatment. Various risk prediction scores for adverse cardiovascular events have been derived from sizable cohorts of patients with IS, including the AS5F [23], STAF [24], PANSCAN [25], CTRAN [26], C_2_HEST [27,28], and LADS [29] scores (Figure 1). These scores integrate information on factors such as age, sex, cardiac history, NIHSS score, biochemical tests, and ECG findings to different extents. Furthermore, these scores have been validated to effectively predict cardiovascular risk in patients with IS. Healthcare providers can stratify patients with IS according to their individual risk profiles and develop tailored prevention strategies. Strong evidence suggests that the AS5F score is an effective tool for assessing AF risk in patients with IS [23]. Using the AS5F score, doctors can promptly screen patients with IS who are at higher risk of AF and take further measures, such as prolonged ECG monitoring, which can help optimize resource utilization. The PANSCAN score is a risk prediction tool used to stratify patients with AIS at high risk of secondary cardiac injury [25]. The study population of this risk prediction model did not include patients with prior history of heart disease or cardioembolic stroke subtype, thereby avoiding confounding factors to a certain extent. The C_2_HEST score, which has been extensively validated, was initially used to predict the risk of AF in Asians without structural heart disease [27], and later tested in Europeans with a prior history of IS [28]. The results indicated that the C_2_HEST score performed well in predicting the risk of AF [28]. The mentioned risk scoring tools are highly practical in clinical settings because they can be obtained from routine medical records and are easy to calculate. However, their sensitivity and specificity have limitations, potentially leading to some high-risk patients being overlooked. Additionally, the applicability of these scoring tools in different populations and regions needs further validation. A systematic review that studied 23 independent cohorts to evaluate such scores, including most of the aforementioned scores, found significant differences in the discriminative ability of these scores for newly detected AF after IS or transient ischemic attack; therefore, their actual effectiveness in guiding clinical decisions remains uncertain [30].

Currently, many hospitals are gradually implementing shift systems, which may affect the continuity of medical records and individualized patient care [31]. This trend highlights the importance of machine learning (ML) and artificial intelligence in the field of cardiac rhythm monitoring and cardiovascular risk prediction post-IS [32]. Although existing risk scoring tools include multiple risk factor parameters, their predictive accuracy still has room for improvement, especially when dealing with increasingly large and complex data [33]. Cho et al. tested the performance of ML models compared to existing cardiovascular risk prediction models and found that ML algorithms, particularly logistic regression, AdaBoost, and neural networks, showed significant advantages in predictive accuracy for cardiovascular risk [34]. Censi et al. developed a risk prediction model that identifies patients with varying risks of atrial fibrillation by quantifying P-wave morphology [35]. Additionally, Ambale-Venkatesh et al. used random forests to predict cardiovascular outcomes, including stroke, AF, and heart failure, showing good accuracy [36].

## 4. Extended Cardiac Rhythm Monitoring to Detect AF after IS

AF prevents the atria from contracting effectively, causing blood to stagnate and pool, particularly in the left atrium. This condition significantly increases the risk of thrombus formation and subsequent embolism to the brain, leading to a cerebral embolism [37]. Since AF is often asymptomatic and intermittent, it may not be detectable by routine ECG or symptom observation in such cases. Therefore, by extending the heart rate monitoring time, the AF burden in AIS patients can be promptly assessed, allowing for appropriate measures to reduce the risk of recurrent stroke and other thromboembolic events [38,39]. At least one-fifth of IS stroke cases are associated with potential cardiac embolic sources [40]. Although 74.4% of significant arrhythmia events occur within the first 24 h after AIS [9], the detection rate of paroxysmal AF increases with a longer duration of cardiac monitoring [41]. Therefore, prolonged cardiac monitoring may detect paroxysmal AF that may not be detected by short-term monitoring, and corresponding anticoagulation treatment plans can be promptly formulated to prevent adverse cardiovascular events.

The European Society of Cardiology recommends screening for arrhythmias in patients with stroke without known AF through 24 h continuous ECG short-term monitoring, followed by long-term (≥72 h) ECG monitoring [42]. In 2019, the American Heart Association/American Stroke Association (AHA/ASA) guidelines proposed that cardiac monitoring within 24 h of stroke onset can more accurately identify potential cardioembolic stroke patients [43]. Furthermore, for patients with cryptogenic stroke, AHA/ASA recommends prolonged cardiac monitoring for at least 30 days to reduce the risk of recurrent stroke [44].

However, currently, there are no clear guidelines regarding the optimal duration and method for post-AIS arrhythmia detection for early AIS management. Randomized controlled trials are necessary to validate the diagnostic effectiveness of different cardiac monitoring methods in patients with IS (Table 1). Noninvasive ambulatory 30-day dynamic ECG monitoring significantly enhanced AF detection rate by more than five times compared with conventional 24 h monitoring [45]. In a study of patients with IS without a history of AF, the incidence of AF within 12 months was higher with implantable ECG monitoring than with conventional 30-day external ECG monitoring [46]. With technological advancements, smartphones are widely used and considered potential devices for long-term cardiac rhythm monitoring to screen for undiagnosed AF [47]. In a multicenter randomized controlled trial, compared to 24 h Holter monitoring, 30-day smartphone-based ECG monitoring showed a higher detection rate of AF and a significant increase in the proportion of patients receiving oral anticoagulation therapy [48]. Additionally, cardiac pacemakers, defibrillators, or other implanted cardiac electronic devices can detect the onset of AF and are important tools for diagnosing and managing cardiac rhythm [49]. Research indicates that in patients with implanted cardiac pacemakers or defibrillators, subclinical atrial tachycardia detected even in the absence of clinical symptoms is associated with an increased risk of subsequent IS [50].

Recent studies have highlighted the potential benefits of extended cardiac monitoring in patients with IS. A systematic review and meta-analysis revealed that prolonged cardiac monitoring was linked to a decreased risk of first-time or recurrent IS, increased detection rate of AF, and a higher likelihood of commencing anticoagulation therapy [51]. Furthermore, a retrospective cohort analysis utilizing claims data from 180,000 American patients with stroke indicated a heightened risk for stroke recurrence among patients who did not receive oral anticoagulation therapy at baseline and had AF first diagnosed >7 days after stroke [52].

**Table 1 healthcare-12-01415-t001:** Published randomized trials evaluating the value of extended cardiac rhythm monitoring in detecting atrial fibrillation after ischemic stroke.

Authors (Year)	Investigational Device	Comparator	Population	No. of Patients	Primary Endpoint
Buck et al. (2021) [46]	Implantable ECG monitoring for 12 months	External ECG monitoring for 30 days	Age ≥ 18 years; IS; excluded those previously documented AF, a pacemaker, an implantable cardioverter-defibrillator, or >7 days of post-stroke external ECG monitoring	300	New AF lasting ≥ 2 min through 12 months
Bernstein et al. (2021) [53]	Insertable cardiac monitors	Usual care	Age ≥ 60 years or age 50–59 years with a documented medical history of at least 1 of the additional stroke risk factors; Stroke attributed to large- or small-vessel disease; absence of AF; no contraindication for OAC	492	Incident AF lasting more than 30 s through 12 months
Wachter et al. (2017) [54]	Enhanced and prolonged monitoring (i.e., 10-day Holter-ECG monitoring at baseline and 3 months and 6 months of follow-up)	Standard care procedures (i.e., at least 24 h of rhythm monitoring).	Age ≥ 60 years; IS; absence of AF; no contraindication for OAC; not suffering from severe ipsilateral carotid or intracranial artery stenosis	398	the occurrence of AF or AFL lasting ≥30 s within 6 months
Gladstone et al. (2014) [45]	30-day event-triggered recorder	24-h Holter monitor	Age ≥ 55 years; cryptogenic IS or TIA within the previous 6 months; absence of AF	572	Newly detected AF lasting ≥ 30 s within 90 days
Higgins et al. (2013) [55]	SP-AM	SP	IS within 7 days; absence of AF or AFL; no contraindication for OAC	100	Detection of AF at 14 days
Koh et al. (2021) [48]	30-day smartphone ECG recording	24-h Holter monitoring	Age ≥ 55 years; absence of AF; IS or TIA within the preceding 12 months	203	AF lasting ≥ 30 s within 3 months

ECG, electrocardiography; IS, ischemic stroke; AF, atrial fibrillation; OAC, oral anticoagulation; AFL, atrial flutter; TIA, transient ischaemic attack; SP-AM, standard practice plus additional monitoring; SP, standard practice.

## 5. Management of Modifiable Risk Factors

According to the EUROASPIRE III survey, about half of the patients with IS do not have sufficient implementation of secondary prevention measures, resulting in the ongoing risk of recurrent stroke and other cardiovascular events [56]. 

### 5.1. Lifestyle Interventions

A randomized controlled trial found that lifestyle interventions, including exercise training, salt restriction, and nutrition advice, can reduce the incidence of recurrent stroke and other vascular events in non-cardioembolic mild IS patients [57]. However, the evidence regarding the effectiveness of lifestyle interventions in reducing cardiovascular event risk after IS seems mixed. A systematic review of 17 randomized controlled trials to evaluate the effect of lifestyle interventions on all-cause mortality and cardiovascular event incidence in patients with IS or transient ischemic attack did not report any significantly favorable results [58].

### 5.2. Exercise Training

Exercise training has been shown to improve outcomes in CVDs [59]. A cohort study involving 441,798 participants with or without CVDs showed that the higher the level of physical activity per week, the lower the risk of mortality [60]. Exercise training may be beneficial in preventing CVDs due to the associated improvement in traditional CVD risk factors, such as type 2 diabetes, hypertension, dyslipidemia, and obesity [61]. In addition, exercise training can also exert cardioprotective effects through various roles such as anti-atherosclerosis [62], reducing inflammation [63], and preventing ischemia-reperfusion injury [64]. AHA/ASA recommends that stroke survivors engage in moderate-to-vigorous physical activity 3–4 times per week for an average duration of 40 min per session, if their physical condition permits [65,66].

### 5.3. Diet

In the PREDIMED study, a Mediterranean diet supplemented with extra-virgin olive oil or nuts was found to be superior to the control diet (with advice on reducing dietary fat) in the primary prevention of CVDs in high-risk individuals [67]. In the CORDIOPREV study, 1002 patients diagnosed with coronary heart disease were randomly assigned to a Mediterranean diet (500 patients) or a low-fat diet (502 patients). The results showed that the incidence of major cardiovascular events in the Mediterranean diet group was significantly lower than that in the low-fat diet group, indicating that the Mediterranean diet is more effective in secondary prevention [68]. These randomized controlled studies highlight the potential of a Mediterranean diet as a primary prevention strategy for CVDs. The benefits of Mediterranean diet on the cardiovascular system may be attributed to the synergistic effects of various nutrients in the diet, which exert their effects on regulation of blood pressure, lipids, oxidative stress, and inflammatory cascades through various mechanisms [69,70].

### 5.4. Smoking Cessation

A study evaluating the impact of smoking reduction and smoking cessation on CVD incidence risk indicated that smoking cessation interventions in adults over 40 years of age could decrease the risk of stroke and MI, whereas reducing smoking has not been consistently significantly associated with reduced risk of CVDs [71]. However, Boulanger et al. used a Poisson regression model to calculate the rate of MI in patients with IS or transient ischemic attack with a history of smoking compared with that in patients without a history of smoking in patients and found no statistically significant difference in their risk of experiencing MI [72]. The combination of behavioral support with pharmacotherapy has been demonstrated to enhance the success of smoking cessation [73,74]. Medications approved by the U.S. Food and Drug Administration for smoking cessation include nicotine replacement therapy products, bupropion, and varenicline [75].

### 5.5. Alcohol and Drug Abuse

For patients with AIS, while low to moderate alcohol consumption has some cardiovascular benefits, drinking more than 60 g per day has been closely linked to stroke recurrence within 90 days and significantly increases the risk of cardiovascular death [76,77,78]. Despite the complex and multidimensional relationship between alcohol consumption and cardiovascular disease, public health authorities clearly discourage both occasional and long-term heavy drinking [79]. A nested case-control study suggested that drug overdose was associated with an increased risk of cardiovascular diseases, particularly arrhythmias, IS, hemorrhagic stroke, and MI [80]. With more states legalizing marijuana, its use, as one of the most commonly abused substances in the U.S., continues to increase [81]. Skipina et al. observed a clear dose–response relationship between increased marijuana use and higher atherosclerotic cardiovascular risk [82]. To mitigate the long-term cardiovascular consequences of drug abuse, comprehensive measures such as psychological counseling, community support, and pharmacological interventions are recommended [83].

### 5.6. Hyperlipidemia

Dyslipidemia has been demonstrated to be an independent predictor of cardiovascular events. Strengthening the prevention and control of dyslipidemia is beneficial for reducing the burden of stroke and MI [84]. In the secondary prevention of stroke, it has been recommended to lower LDL-C levels to below 70 mg/dL to prevent major cardiovascular events [65]. A meta-analysis of 27 randomized trials (involving 174,149 participants) demonstrated that statin therapy, which lowers low-density lipoprotein cholesterol levels, leads to a reduced risk of major coronary artery events [85]. Therefore, statin therapy can be used to treat patients at higher risk of vascular events and not just to treat dyslipidemia. A follow-up study of 4731 patients with stroke or transient ischemic attack, without known coronary heart disease, found that administration of atorvastatin 80 mg/day can significantly reduce the risk of coronary heart disease in these patients, regardless of the baseline stroke subtype [86]. Proprotein convertase subtilisin/kexin type 9 (PCSK9) inhibitors and ezetimibe may also reduce the cardiovascular risk of IS. Recent meta-analyses have shown that the addition of PCSK9 inhibitors or ezetimibe to statin therapy can reduce the incidence of nonfatal MI and stroke in adults with very high or high cardiovascular risk; however, the cardiovascular benefit of this type of therapy has not been demonstrated in individuals with intermediate or low cardiovascular risk [87]. Icosapent ethyl is a highly purified eicosapentaenoic acid ethyl ester that has been shown to lower triglyceride levels [88]. The REDUCE-IT trial demonstrated the effectiveness and safety of eicosapentaenoic acid ethyl ester (4 g per day) in reducing the risk for stroke and cardiovascular mortality [89]. Lipid levels should be measured 4 to 12 weeks after starting LDL-C lowering medications to evaluate patient response and lifestyle adherence, then repeated every 3 to 12 months [90].

### 5.7. Hyperglycemia

Elevated blood glucose upon hospital admission increases in-hospital mortality risk and poor functional recovery in patients with IS, especially in those without a history of diabetes [91]. Fasting plasma glucose or hemoglobin A1c (HbA1c) should be used to evaluate the glycemic status of all patients with IS [92]. For patients with AIS, continuous blood glucose monitoring for at least 72 h post-stroke is necessary, regardless of diabetes status [93]. Nevertheless, studies have not found significant clinical benefits in strict blood glucose control for AIS patients, and it may even increase the risk of hypoglycemia [94,95,96]. A meta-analysis of potential risk factors for MI following IS/transient ischemic attack episodes indicated that there was no significant correlation between a history of diabetes and MI [72]. Therefore, blood glucose control during the acute phase of stroke is relatively relaxed [43,97]. For patients with AIS with blood glucose levels > 10.0 mmol/L (180 mg/dL), it is appropriate to prioritize intravenous insulin administration to maintain blood glucose levels within the range of 7.8–10.0 mmol/L (140–180 mg/dL) [43]. However, considering the increased risk of cardiovascular complications with long-term hyperglycemia, long-term blood glucose control should be performed for patients with IS. A target of HbA1c control of less than 7% is considered reasonable [92,98]. Randomized controlled trials evaluating glucagon-like peptide 1 receptor agonists (such as liraglutide or semaglutide) have shown that these agents can lower the occurrence of cardiovascular events in patients with type 2 diabetes, particularly in individuals with diagnosed CVDs or those at high risk of CVDs [99,100].

### 5.8. Hypertension

A meta-analysis including 58 studies (131,299 stroke participants) showed that patients with IS with a history of hypertension have a significantly increased risk for MI [72]. Recent guidelines recommend that hypertensive treatment should be administered to AIS patients with a blood pressure of higher than 140/90 mmHg, after their neurological function has stabilized [101]. It may be reasonable to control blood pressure at 130/80 mmHg or lower, which can help reduce the risk of complications such as thrombosis and bleeding in patients with stroke [101]. Patients typically need repeated blood pressure assessments within the first month of treatment to ensure therapy effectiveness. If blood pressure targets are met, subsequent assessments are usually conducted every 3 to 6 months [101]. ACE inhibitors can be used as first-line treatment for patients with IS, as they can prevent recurrent stroke and reduce the risk of other CVDs such as stable ischemic heart disease, AF, and heart failure [101]. In the Perindopril Protection Against Recurrent Stroke Study, 6105 patients with a history of transient ischemic attack or stroke were randomly assigned to receive intensified blood pressure control (3051 patients) or placebo treatment (3054 patients). Over a 4-year period, it was observed that intensified blood pressure control reduced the relative risk of stroke by 28%, and the overall risk of major vascular events by 26% [102].

## 6. Antiplatelet and Anticoagulant Therapy

The AHA/ASA states that the use of antiplatelet drugs remains the recommended approach for preventing recurrent stroke and other cardiovascular events in most patients with non-cardioembolic IS [43,44]. For patients with non-cardioembolic IS, dual antiplatelet therapy with aspirin plus clopidogrel or ticagrelor may be beneficial [44]. A large international randomized controlled trial found that among patients with mild IS (NIHSS score ≤ 3), aspirin plus clopidogrel therapy reduced the risk of composite outcomes of IS, MI, or vascular death compared with aspirin monotherapy but increased the risk of major bleeding [103]. Another trial focusing on patients with mild-to-moderate acute non-cardioembolic IS (NIHSS score ≤ 5) demonstrated that combination therapy involving aspirin and ticagrelor was linked to a lower risk of recurrent stroke or death within 30 days but a higher risk for major bleeding [104].

For patients with AIS related to AF, oral anticoagulants, including non-vitamin K antagonist oral anticoagulants and vitamin K antagonists such as warfarin, are recommended [42]. Patients with IS continue to face a higher risk of bleeding even after taking anticoagulants, especially older patients and those with other underlying conditions. Therefore, exploring new treatments to balance the prevention of recurrent stroke and the occurrence of secondary bleeding is necessary. Asundexian, a small molecule inhibitor of human activated coagulation factor XI (FXIa), is a widely studied new drug that can inhibit thrombosis without increasing the risk of bleeding [105]. In a recent randomized controlled trial, it was found that asundexian (20 mg or 50 mg per day) was more effective than the standard dose of apixaban in reducing bleeding rates in patients with AF [106]. However, in patients with acute non-cardioembolic IS, asundexian was not significantly different from placebo in reducing composite outcomes of covert brain infarcts or IS, or composite outcomes of major or clinically relevant non-major bleeding [107]. Further research is required to assess whether the FXIa inhibitor asundexian can be considered a safe and effective treatment option for patients with IS.

The initiation or resumption of oral anticoagulation after IS depends on the extent of brain imaging and neurological deficit [108]. If it is anticipated that the size of the infarction will not significantly increase the risk for secondary hemorrhagic transformation and there are no contraindications, anticoagulation therapy can be started [108]. A study of 395 patients with AIS with AF compared the effects of different anticoagulation initiation times (within 4 days, between days 5–14, and no anticoagulation within 14 days after stroke onset) on functional outcomes, recurrent ischemic events, and bleeding complications. The result showed that the optimal time to initiate anticoagulation treatment may be between 5 and 14 days after IS onset [109].

## 7. Optimization of Early Recovery Care

A significant challenge within global healthcare systems is ensuring continued medical care for post-discharge patients with stroke. The transition from hospital to home often results in the discontinuation of continuous medical supervision, which is a critical gap that must be addressed [110,111]. Promoting the continuity of medical care and early implementation of rehabilitative care and rehabilitation measures can accelerate patient recovery and reduce the risk of recurrent stroke and cardiovascular complications.

### 7.1. Assessment of the Quality of Nursing Services

Relying solely on family members to provide high-quality care for patients with IS is inadequate. Medical institutions should provide nursing staff appointment services before patient discharge and coordinate the hiring of care teams after discharge. Medical institutions need to train nursing personnel in various skills and roles to improve their ability to assess and address the needs of patients with stroke [112]. Nursing teams should prioritize personalized care plans that align with the goals, values, and lifestyles of patients with IS. Establishing clear performance assessment indicators and utilizing standardized scoring tools are essential to evaluate the progress and effectiveness of care plans [113]. This patient-centered approach can improve treatment compliance, reduce healthcare disparities, enhance patient well-being, and drive continuous quality improvement and innovation in care delivery. Nursing personnel may experience health issues during the caregiving process, which can negatively affect the prognosis of patients with stroke over time [114,115]. Therefore, nursing staff should not overlook self-care [112].

### 7.2. Multidisciplinary Remote Access

Due to the many similarities in the risk factors for IS and CVD, multidisciplinary nursing approaches are often used in the management of IS. Cardiologists play an important role in the treatment of IS. Collaboration with cardiologists can provide more comprehensive and personalized risk management and treatment for patients with stroke, as well as support for rehabilitation and prevention of cardiovascular complications [116]. Post-stroke telehealth care integrates customized information charts, home remote monitoring, and multidisciplinary team video visits. This model enhances the care transition for post-stroke patients in remote areas by promoting collaboration among nurses, pharmacists, and doctors [117,118]. The process includes customizing the information charts of key post-discharge parameters, covering neurological function, brain imaging, ECG, blood pressure, blood glucose, lipids, and cardiac rhythm and function, and sharing them during the first telehealth visit. During the first 1 to 2 weeks post-discharge, primary care nurses would assess the patient’s general condition and recovery progress. Between 4 and 8 weeks, pharmacists would review medication effectiveness, adherence, and side effects. Between 6 and 12 weeks, doctors would conduct a comprehensive health assessment and adjust the treatment plan or offer advice as needed [118]. A meta-analysis indicated that stroke-specialized multidisciplinary team nursing was more effective than conventional nursing approaches [119]. In cases where patients with stroke exhibit suspected or newly diagnosed cardiovascular complications at primary healthcare centers, a referral to the nearest tertiary care hospital is warranted for comprehensive evaluation, diagnosis, and management.

### 7.3. Clinical Information Sharing

In acute care settings, emergency physicians typically make prompt decisions and interventions, while having little access to the patient’s historical clinical information. Although access to a comprehensive history and past clinical data can greatly assist in the development of diagnosis and treatment plans, it is not always available in emergency situations. The pilot study conducted by Overhage et al. provided the first evidence of the feasibility of sharing clinical information across diverse healthcare systems [120]. Traditional paper-based storage systems do not support information sharing among different institutions. The development of electronic medical records systems that can be accessed by different institutions within the healthcare system would be highly beneficial [121]. For patients, the greatest advantage of cross-department electronic medical record systems is the reduction in repeated tests and treatments (266/501, 53.1%); for physicians, this system can facilitate timely access to patients’ medical records (314/409, 76.8%) and promote continuity of care (259/409, 63.3%) [122]. Despite the evident benefits, we have observed a low level of information-sharing system utilization. For example, only 2.3% (6142/271,305) of patients in the emergency department had access to the Health Information Exchange system [123]. This low utilization hinders access to patients’ medical records and disrupts the continuity of nursing care.

## 8. Conclusions and Future Directions

CVD risk is always present in patients with IS and significantly impacts their prognosis and risk of death. Although the currently available risk prediction scoring tools have some predictive value for adverse cardiovascular events after stroke, their suitability in clinical practice remains uncertain. Further validation and improvement are needed to determine their sensitivity and specificity. Therefore, promoting the use of advanced technologies such as artificial intelligence or machine learning to analyze extensive patient data, identify new risk factors, and further refine risk prediction models may potentially enhance accurate risk assessment and guidance for individualized prevention and treatment. While antiplatelet and anticoagulant therapies are common treatments for improving cardiovascular outcomes after IS, they may also carry risks and side effects. In the future, there is a need to explore new therapeutic avenues and evaluate their long-term efficacy and feasibility. Furthermore, the strategies discussed in this article to promote continuity of care across settings are limited to the systemic level. To ensure safe and effective transitions between care settings for patients with stroke, concerted efforts are needed at societal and individual levels.

## Figures and Tables

**Figure 1 healthcare-12-01415-f001:**
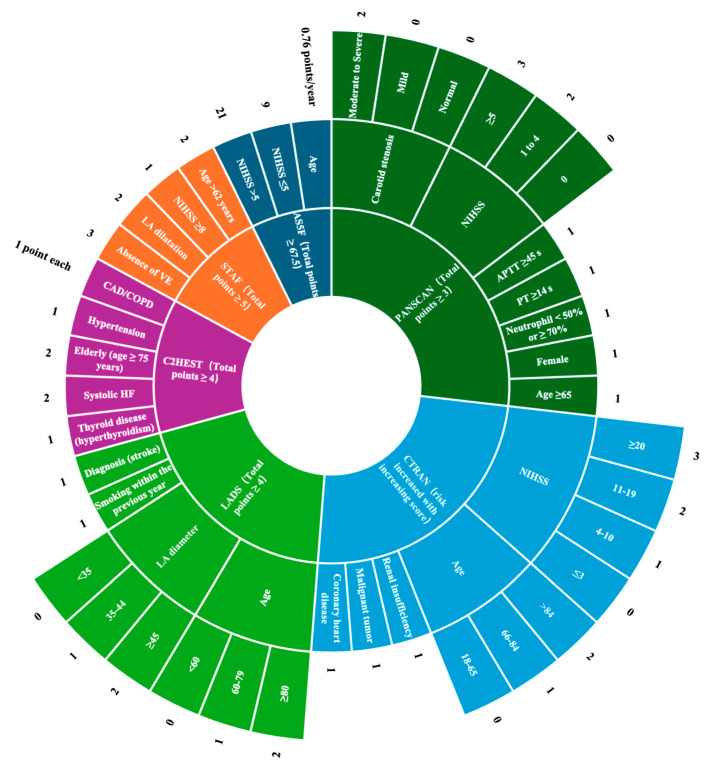
Cardiovascular risk factors and parameters after ischemic stroke. HF, heart failure; PT, prothrombin time; APTT, activated partial thromboplastin time; LA, left atrial diameter; VE, vascular etiology; CAD, coronary artery disease; COPD, chronic obstructive pulmonary disease; NIHSS, national institutes of health stroke scale.

## Data Availability

Not applicable.

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
