# Peer review of "Post-Ischemic Stroke Cardiovascular Risk Prevention and Management"

_healthcare, 2024, doi:10.3390/healthcare12141415_

Round 1

Reviewer 1 Report

Comments and Suggestions for Authors

I thank the authors for preparing a nice and clear manuscript on prevention of the cardiovascular events in the post-stroke period. The text is well-written and the material is presented in a logical way. The only thing that is missing,  to my point, is a more detailed description on the mechanisms underlying the increased risk of cardiovascular events and their relations to the stroke subtypes. And a discussion of what is the cause and what is the sequence. When can one talk about stroke as a factor increasing the risk of CVD and when stroke is just an indicator of a pre-existing  but previously undefined cardiac deficits? Are their any clinical markers (ECG, MRI) that may point to a particular stroke subtype (for example, see Kulesh et al.,  2022 https://nnp.ima-press.net/nnp/article/view/1779) and help to differentiate between these cases?

Author Response

Comments: The only thing that is missing,  to my point, is a more detailed description on the mechanisms underlying the increased risk of cardiovascular events and their relations to the stroke subtypes. And a discussion of what is the cause and what is the sequence. When can one talk about stroke as a factor increasing the risk of CVD and when stroke is just an indicator of a pre-existing  but previously undefined cardiac deficits? Are their any clinical markers (ECG, MRI) that may point to a particular stroke subtype (for example, see Kulesh et al.,  2022 https://nnp.ima-press.net/nnp/article/view/1779) and help to differentiate between these cases?
Response: Thank you for pointing this out. We agree with this comment. Therefore, we have added corresponding content to the paper. The concept of stroke-heart syndrome implies that cardiac dysfunction occurs after the onset of neurological deficits. AIS can cause cardiac dysfunction through pathways such as sympathetic and parasympathetic imbalance, catecholamine surge, immune and inflammatory responses, and gut micro-biota dysbiosis. When stroke acts as an independent risk factor for CVD, it may, along with other risk factors such as hyperlipidemia, hyperglycemia, and hypertension, increase the risk of recurrent stroke and cardiovascular events. Stroke may also indicate underlying cardiac defects or cardiovascular issues such as valvular heart disease or AF, which may underlie or coexist with stroke. Insular cortex lesions can help differentiate cardioembolic stroke (caused by AF) from non-cardioembolic stroke (caused by atherosclerosis or cerebral microvascular disease) and are considered potential neuroimaging markers for identi-fying IS subtypes due to cardiac embolism. These changes can be found in the revised manuscript, respectively - page 1, paragraph 2, line 44, and page 2, paragraph 4, line 82.

Reviewer 2 Report

Comments and Suggestions for Authors

The article Post-Ischemic Stroke Cardiovascular Risk Prevention and Management highlights the main points related to preventing cardiovascular complications after stroke.

Regarding risk factors, it is worth mentioning:

1 - alcohol abuse

2 - abuse of medicines and drugs.

It should also be mentioned that one of the cardiovascular complications may be venous thromboembolism.

Author Response

Comment 1: Regarding risk factors, it is worth mentioning: 1 - alcohol abuse  2 - abuse of medicines and drugs.
Response 1: Thank you for pointing this out. We agree with this comment. Therefore, we have added strategies for managing the risk factors of alcohol abuse and abuse of medicines and drugs to the paper. This change can be found in the revised manuscript - page 8, paragraph 2, line 271.

Comment 2: It should also be mentioned that one of the cardiovascular complications may be venous thromboembolism.
Response 2: We agree with this comment. Therefore, to emphasize this point, we add the epidemiological characteristics of venous thromboembolism complicating ischemic stroke. This change can be found in the revised manuscript - page 2, paragraph 3, line 70.

Reviewer 3 Report

Comments and Suggestions for Authors

Basing on  analysis of cardiovascular events in large cohorts of patients including the ASSF, STAF, PANSCAN, CTRAN, C2HEST and LADS scores they conclude that the ASSF score effectively predicts the cardiovascular risk in patients with ischemic shock. As a whole the manuscript is well written and refers to essential questions, however I have some critical comments:

1. Figure 1 is not informative and should be replaced by another one showing links between the risk factors and analyzed parameters.

2. Discussing the role of exercise training, diet, smoking cessation, hyperlipidemia, hyperglycemia and hypertension, the authors should give information how long the hyperlipidemia, hyperglycemia and hypertension were monitored and how long exercise training and smoking cessation were continued.

3. The item 7 “ Optimization of early recovery care” refers to observations that are not directly related to the topic of this review. Basing on their analysis, the authors should more explicit propose how the early recovery care should be organized and what parameters should be monitored. They  should omit vague statements which are not directly related to the subject of the review.

Comments on the Quality of English Language

Quality of English Language is satisfactory, although some sentences are too long and intricated. Professional language coorection would be desirable at the final stage. 

Author Response

Comment 1: Figure 1 is not informative and should be replaced by another one showing links between the risk factors and analyzed parameters.
Response 1: Thank you for pointing this out. We agree with this comment. Therefore, we replaced it with another figure showing cardiovascular risk factors and their parameters after ischemic stroke. This figure can be found in the revised manuscript - page 4, line 148.

Comment 2: Discussing the role of exercise training, diet, smoking cessation, hyperlipidemia, hyperglycemia and hypertension, the authors should give information how long the hyperlipidemia, hyperglycemia and hypertension were monitored and how long exercise training and smoking cessation were continued.
Response 2: We agree with this comment.Monitoring lipid levels, blood glucose, and blood pressure, along with implementing exercise training and smoking cessation, are long-term processes after an ischemic stroke to maintain cardiovascular health and reduce recurrent stroke risk. Therefore, to emphasize this point, we have added the time to monitor hyperlipidemia, hyperglycemia, and hypertension after ischemic stroke. These changes can be found in the revised manuscript, respectively - page 8, paragraph 3, line 307; page 9, paragraph 2, line 314; and page 9, paragraph 3, line 338.

Comment 3: The item 7 “ Optimization of early recovery care” refers to observations that are not directly related to the topic of this review. Basing on their analysis, the authors should more explicit propose how the early recovery care should be organized and what parameters should be monitored. They should omit vague statements which are not directly related to the subject of the review.
Response 3: Thank you for pointing this out. We agree with this comment. Therefore, we have added specific processes for early medical care after stroke and omitted vague statements that were not relevant to the topic. This change can be found in the revised manuscript - page 11, paragraph 1, line 413.

Reviewer 4 Report

Comments and Suggestions for Authors

The article entitled "Post-Ischemic Stroke Cardiovascular Risk Prevention and Management" is well written, well documented and as the authors mentioned, most studies don't focus on preventive management strategies for post ischemic stroke heart disease. However, some minor revision are needed to be made:

1. How the authors made the research for this manuscris? Where they searched, what were the key words, how many articled did they find and how did they exclude some articles.

2. The authors mentioned about many risk scores that are already published. What is their personal opinion about them? They mentioned just about one systematic review with their utility.

3. About subsection "extended cardiac rhythm monitoring to detect AF after IS" - there is just a few words written about loop recorded, I think there are many more articles in the literature about its importance. Moreover, there are some devices that the patients can already have for other pathologies (e.g. pacemakers/defibrilators) that can have diagnostic tools for AF. The authors should mention about them and about their importance. 

4. The authors just mentioned about AI and machine learning as a future perspective. I suggest to extend this section and talk more about what already exist. 

Comments on the Quality of English Language

Minor English revision. Some sentences are too long and lose interest.

Author Response

Comment 1: How the authors made the research for this manuscris? Where they searched, what were the key words, how many articled did they find and how did they exclude some articles.
Response 1: We searched using medical literature databases such as PubMed, Embase, and the Cochrane Library with key search terms including, but not limited to, the following: ischemic stroke, cardiovascular disease, adverse cardiovascular events, risk management, risk stratification, cardiac rhythm monitoring, primary prevention, secondary prevention, lifestyle interventions, exercise training, diet, smoking cessation, hyperlipidemia, hyperglycemia, hypertension, antiplatelet and anticoagulant therapy, early recovery care. Since this article is a review, not a systematic review or Meta-analysis, it requires further addition of specific risk factors or management strategy keywords as needed. We will start by searching relevant literature using keywords in databases, followed by initial screening based on titles and abstracts. We will then read the full texts of articles that fit the review theme, organize, analyze, and summarize the viewpoints, evidence, and findings of the existing literature to provide a solid foundation for writing the article.

Comment 2: The authors mentioned about many risk scores that are already published. What is their personal opinion about them? They mentioned just about one systematic review with their utility.
Response 2: Thank you for pointing this out. We agree with this comment. Therefore, we have added to the discussion of the strengths and weaknesses of these risk scores. This change can be found in the revised manuscript - page 3, paragraph 3, line 139.

Comment 3: About subsection "extended cardiac rhythm monitoring to detect AF after IS" - there is just a few words written about loop recorded, I think there are many more articles in the literature about its importance. Moreover, there are some devices that the patients can already have for other pathologies (e.g. pacemakers/defibrilators) that can have diagnostic tools for AF. The authors should mention about them and about their importance.
Response 3: Thank you for pointing this out. We agree with this comment. Therefore, to emphasize the importance of prolonged heart rate monitoring, we have added the appropriate content on page 4, paragraph 2, line 167. Additionally, cardiac pacemakers, defibrillators, or other implanted cardiac electronic devices can detect the onset of atrial fibrillation and are important tools for diagnosing and managing cardiac rhythm. This change can be found in the revised manuscript - page 5, paragraph 3, line 200.

Comment 4: The authors just mentioned about AI and machine learning as a future perspective. I suggest to extend this section and talk more about what already exist.
Response 4: We agree with this comment. Therefore, we have extended this section to discuss current applications of machine learning and artificial intelligence in the field of cardiac rhythm monitoring and cardiovascular risk prediction after ischemic stroke. This change can be found in the revised manuscript - page 4, paragraph 1, line 153.